# RoBERTa-Based Keyword Extraction from Small Number of Korean Documents

**So-Eon Kim** [1][ID]**, Jun-Beom Lee** [1][ID]**, Gyu-Min Park** [1][ID]**, Seok-Man Sohn** [2] **and Seong-Bae Park** [1,*][ID]

[1] School of Computing, Kyung Hee University, Yongin 17104, Republic of Korea; sekim0211@khu.ac.kr (S.-E.K.); jblee9410@khu.ac.kr (J.-B.L.); pgm1219@khu.ac.kr (G.-M.P.)
[2] Korea Electric Power Research Institute, Daejeon 34056, Republic of Korea; happysohn@kepco.co.kr
* Correspondence: sbpark71@khu.ac.kr

**Abstract:** Keyword extraction is the task of identifying essential words in a lengthy document. This process is primarily executed through supervised keyword extraction. In instances where the dataset is limited in size, a classification-based approach is typically employed. Therefore, this paper introduces a novel keyword extractor based on a classification approach. The proposed keyword extractor comprises three key components: RoBERTa, a keyword estimator, and a decision rule. RoBERTa encodes an input document, the keyword estimator calculates the probability of each token in the document becoming a keyword, and the decision rule ultimately determines whether each token is a keyword based on these probabilities. However, training the proposed model with a small dataset presents two challenges. One problem is the case that all tokens in the documents are not a keyword, and the other problem is that a single word can be composed of keyword tokens and non-keyword tokens. Two novel heuristics are thus proposed to tackle these problems. To address these issues, two novel heuristics are proposed. These heuristics have been extensively tested through experiments, demonstrating that the proposed keyword extractor surpasses both the generation-based approach and the vanilla RoBERTa in environments with limited data. The efficacy of the heuristics is further validated through an ablation study. In summary, the proposed heuristics have proven to be effective in developing a supervised keyword extractor with a small dataset.

**Keywords:** keyword extraction; sequence labeling; post-processing; RoBERTa; learning with small dataset



## 1. Introduction

Keyword extraction is a task of identifying keywords in a lengthy document, and it is considered one of the essential tasks in data analysis, data mining, and information retrieval [1–4]. In the current environment, where countless documents are published through various media [5,6], it becomes time-consuming for humans to process all documents and identify their keywords. Consequently, the demand for automatic keyword extraction has increased [7–9].

Automatic keyword extraction has been addressed using unsupervised and supervised methods [10]. Two representative examples of unsupervised keyword extraction are Tf-idf [11] and PageRank [12,13]. Tf-idf detects keywords based on word frequency [14–16], while PageRank utilizes graph-based features [17–19]. However, these methods, lacking an understanding of the overall semantic information of a document, generally achieve relatively low accuracy. To address this issue, supervised keyword extraction has been introduced, reporting high performance [20]. Nevertheless, these methods typically necessitate a large amount of training data, posing a challenge when only limited data are available.

Supervised keyword extraction has been studied through two approaches: the classification-based approach and the generation-based approach. The classification-based approach involves extracting keywords from a document by evaluating every token in the document to determine if it is a keyword or not [21–23]. In contrast, the generation-based

approach uses a generative language model to abstractively generate keywords for an input document [24,25]. According to numerous studies [25], generative language models like BART [26] outperform classification-based methods in extraction accuracy, making the generation-based approach more commonly adopted for keyword extraction. However, a drawback of generative language models is their requirement of a large amount of training data, surpassing that needed by classification-based methods in general.

This paper proposes a novel keyword extractor designed to be trained with a small number of Korean documents. The extractor addresses three problems that are challenging for generative language models. Firstly, due to the unavailability of large-scale keyword-spotting documents for minor languages such as Korean, the extractor must be trained with a limited amount of data. Consequently, the proposed keyword extractor adopts the classification-based approach [27] instead of the generation-based approach, utilizing RoBERTa [28], a BERT-based classifier, as a base model due to its outstanding performance among classification-based methods.

The second problem arises from RoBERTa determining whether every token in a document is a keyword or not. Consequently, it is highly possible that RoBERTa may not produce any keywords for a document since it is a token-level classifier. To address this, a heuristic is employed to identify the word to which the token with the highest probability belongs as a keyword. Additionally, since every word is expressed as a sequence of tokens in RoBERTa, owing to its BPE tokenizer, even a single word can be composed of multiple tokens with varying keyword predictions. The third problem is to unify the keyword predictions of multiple tokens for a single word. To address this issue, a heuristic is applied, considering a word as a keyword only when all its tokens are predicted as keywords, leveraging RoBERTa's reliability in keyword extraction.

Intensive experiments are conducted using a dataset of Korean Power Plant Outage Reports. Although the dataset is small, the experiments demonstrate that RoBERTa achieves high performance compared to BART, a generative language model. Moreover, it is evident that vanilla RoBERTa faces the second and third problems, which the proposed heuristics effectively solve. Particularly, the proposed keyword extractor achieves an average 6.4 and 12.8 higher BLEU and accuracy, respectively, compared to vanilla RoBERTa. Furthermore, it outperforms vanilla RoBERTa even in human evaluation, affirming the efficacy of the proposed keyword extractor for datasets with a limited number of training instances.

The main contributions of this work are

- This paper analyzes problems that may occur when a supervised classification-based keyword extractor is trained with a small number of data.
- This paper proposes two heuristics for sophisticated keyword extraction to solve problems of classification-based keyword extractors trained with a small number of data.
- The proposed keyword extractor outperforms not only the generation-based keyword extractor but also the classification-based keyword extractor in the environment of a small number of data.

The rest of this paper is organized as follows. Section 2 introduces the previous studies on automatic keyword extraction. Section 3 describes two problems in learning a RoBERTa-based keyword extractor with a small number of documents and addresses how to tackle these problems, and Section 4 provides the experimental results. Finally, Section 5 indicates the conclusions and future work.

## 2. Related Work

Traditionally, keyword extraction has been studied in an unsupervised way. Tf-idf [11,29] is a representative unsupervised method, where it detects keywords statistically based on the frequency of the tokens in a document. The methods of extracting keywords using term frequency like tf-idf require a large corpus to improve their performance. Thus, efforts to use not only term frequency but also some other features have been undertaken in order to solve this problem. KP-Miner utilized word length and positional information in

addition to term frequency [30], RAKE considered the frequency of other tokens that a term co-occurs with [31], and YAKE calculated a keyword score with the term frequency and some hand-crafted features [8].

Graph-based keyword extraction is another way to extract keywords unsupervisedly with purely statistical information. TextRank [12,32] is a representative graph-based approach, where it builds an undirected, unweighted word graph representation and then finds out the keywords using the PageRank algorithm. On the other hand, SGRank [33] uses some manually crafted features in addition to TextRank, where the features are designed to complement TextRank since some critical information for keyword extraction cannot be captured by a graph-based method. KeyphraseDS [34] filters out non-keywords from the keyword candidates before TextRank using the similarity of a candidate to a document topic to keep stopwords from becoming keywords. These unsupervised methods have an advantage of requiring small data and faster inference, but their accuracy is relatively low because they extract keywords without a semantic understanding of a document. For this reason, supervised-based methods that extract keywords grasping the semantics of a document have begun to attract the attention of the keyword extraction research community.

Supervised keyword extraction is further divided into a classification-based approach and a generation-based approach. The classification-based approach is also called a sequence labeling approach. That is, it aims to determine whether each input token in a document is a keyword or not. For instance, Luan et al. [35] determined keywords with part-of-speech embeddings and the combinatorial features of words and characters. On the other hand, Chun et al. [36] proposed a novel embedding for sequence labeling, which integrates character-level embedding and word-level embedding to find the most intuitive character or word in a document. However, such classification-based methods always have a possibility that all tokens in a document are determined to be non-keywords.

Many studies have attempted to solve the problems of the classification-based approach by adopting a generative model based on a sequence-to-sequence architecture. This is because the generation-based approach generates keywords abstractly from a document and thus it outputs some keywords anyway. In the early studies, RNN [37] was used as a dominant keyword generator. However, many attempts have been made to improve keyword extraction performance recently by adopting a generative language model. For example, BART [26], a transformer-based language model, showed a higher performance than RNN [25]. Recently, there have also been efforts to generate keywords using prompts [38]. However, the main drawback of this generation-based approach is that it requires a great volume of data for its training. As a result, this paper takes the classification-based approach to solve keyword extraction for a dataset with a small number of documents.

## 3. Keyword Extraction with Small Number of Documents

### 3.1. Supervised Keyword Extraction

Keyword extraction is a task to extract the most relevant words from a document $\mathbf{W} = [w_1, \ldots, w_m]$, a sequence of word $w_i$'s. This task can be solved in various ways. One way is the generation-based approach in which the keywords for an input document $\mathbf{W}$ are generated by a generative language model. That is, a generative language model generates $\mathbf{Y} = [\hat{y}_1, \ldots, \hat{y}_o]$, where $o$ is the targeted number of keywords. One benefit of this approach is that it can generate even the words that do not appear in $\mathbf{W}$ as keywords. However, it has a disadvantage of requiring a large amount of data to train the generative language model.

The other way to keyword extraction is the classification-based approach. This approach is usually formulated as sequence labeling. That is, keyword extraction in this approach can be considered as a series of word-level classifications. Under this approach, the keywords for $\mathbf{W}$ are extracted by determining whether every $w_i \in \mathbf{W}$ is a keyword or not. In order to train a word-level classifier, a sequence of golden labels $\mathbf{Y} = [y_1, \ldots, y_m]$ for each $\mathbf{W}$ should be given where $|\mathbf{X}| = |\mathbf{Y}|$. In addition, all $y_i$'s are boolean so that $y_i$ is `keyword` if the corresponding $w_i$ is a keyword and `non-keyword` otherwise. Formally, let $f(\cdot)$ be a word-level classifier trained with a dataset $D = \{(\mathbf{W}_i, \mathbf{Y}_i)\}_{i=1}^{N}$. Then, when a new

document $\bar{\mathbf{W}} = [\bar{w}_1, \ldots, \bar{w}_m]$ is given, its label $\bar{\mathbf{Y}} = [\bar{y}_1, \ldots, \bar{y}_m]$ is determined by applying $f(\cdot)$ to every $\bar{w}_i \in \bar{\mathbf{W}}$. That is, $\bar{y}_i$ is obtained by

$$\bar{y}_i = f(\bar{w}_i; \bar{\mathbf{W}}). \tag{1}$$

Here, $\bar{\mathbf{W}}$ is used as a context for this classification.

This paper adopts a RoBERTa-based model as the classifier $f(\cdot)$ in Equation (1) since RoBERTa achieves good performances in many sequence labeling tasks [39,40]. The structure of the proposed classifier is depicted in Figure 1. It consists of three components of RoBERTa, a keyword estimator, and a decision rule. The input document $\mathbf{W}$ is first encoded as vectors by RoBERTa. Since RoBERTa is a BERT-based model, every word in $\mathbf{W}$ is further divided into a few tokens by the byte-level BPE, a standard RoBERTa tokenizer. Thus, the document $\mathbf{W} = [w_1, \ldots, w_m]$ is represented as a sequence of tokens $\mathbf{X} = [x_1, \ldots, x_n]$ where $n > m$. Then, the task of keyword extraction becomes predicting $\mathbf{Y} = [y_1, \ldots, y_n]$ from $\mathbf{X}$. Following the standard RoBERTa application, the `[cls]` token is assumed to be in $\mathbf{X}$ and is regarded as $x_0$.

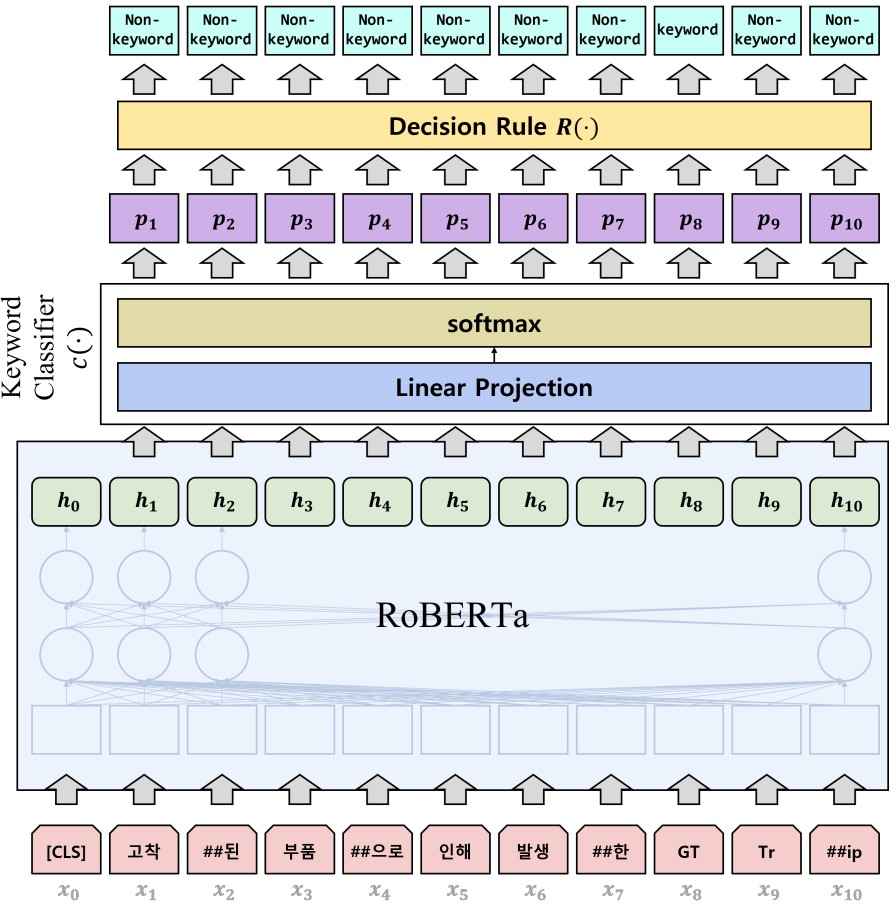

**Figure 1.** Overall architecture of sequence-labeling-based keyword extraction using RoBERTa. The example sentence is "GT (Gas Turbine) trip caused by stuck parts..." in Korean, and the "##" in some tokens implies that it is a token that is connected to a previous token without a white space. In this example, 'GT' is a keyword because the keyword extraction domain is about finding the objects causing an outage or a failure.

For every $x_i \in \mathbf{X}$, a corresponding vector representation $\mathbf{h}_i$ is obtained by

$$\mathbf{h}_i = RoBERTa(x_i; \mathbf{X}) \in \mathbb{R}^E,$$

where $E$ is the embedding size. Note that RoBERTa is encoder-centric since it is based on BERT. Thus, an additional keyword estimator is required to adopt RoBERTa in an end-to-end style [22,23]. This paper uses a linear projection with softmax as the keyword estimator $c(\cdot)$ where it takes $\mathbf{h}_i$ as an input and calculates the probability of $\mathbf{h}_i$'s being a keyword. That is,

$$c(\mathbf{h}_i) = \text{softmax}(\mathbf{h}_i \mathbf{W} + \mathbf{B}), \tag{2}$$

where $\mathbf{W} \in \mathbb{R}^{E \times C}$ and $\mathbf{B} \in \mathbb{R}^{n \times C}$ are learnable parameters and $C$ is the number of class labels. Since $\bar{y}_i$ in Equation (1) can have only two values of `keyword` and `non-keyword`, $C$ is two in this paper. This implies that the number of nodes in the softmax of $c(\cdot)$ is also two. Thus, $c(\mathbf{h}_i)$ in Equation (2) returns $\mathbf{p}_i = \langle p_i^T, p_i^F \rangle$, a vector of two probabilities. $p_i^T$ is the probability that $x_i$ becomes a keyword and $p_i^F$ is the probability that $x_i$ is not a keyword. As a result, $p_i^F = 1 - p_i^T$ holds. In a word,

$$\begin{aligned} \mathbf{p}_i &= c(\mathbf{h}_i) \\ &= c(RoBERTa(x_i; \mathbf{X})). \end{aligned} \tag{3}$$

The final keyword prediction is then made from $\mathbf{p}_i$ by a simple rule

$$R(\mathbf{p}_i) = \begin{cases} \texttt{keyword} & \text{if } p_i^T \geq p_i^F, \\ \texttt{non-keyword} & \text{otherwise.} \end{cases} \tag{4}$$

Therefore, $y_i$ is determined by

$$\begin{aligned} y_i &= f(x_i; \mathbf{X}) \\ &= R(c(RoBERTa(x_i; \mathbf{X}))). \end{aligned} \tag{5}$$

The parameters of $\mathbf{W}$ and $\mathbf{B}$ of the keyword estimator $c(\cdot)$ in Equation (2) are optimized with the training dataset $D$. RoBERTa is also fine-tuned with $D$.

### 3.2. Sequence Labeling for Small Number of Documents

Note that the size of the training data $D$ should be large to obtain a good performance with the proposed keyword extractor in Equation (5). However, minor languages such as Korean usually do not have a large dataset labeled for keyword extraction. If the size of $D$ is very small, the keyword extractor can suffer from two kinds of problems. One problem is that the keyword extractor is underfitted to the training data $D$ and shows a deteriorated performance. One possible phenomenon by the underfitted keyword extractor is that all words in a document are predicted to be non-keywords since there are usually many more non-keywords than keywords in a document. The other problem is that a single word can be predicted to be both a keyword and a non-keyword. This is because the keyword extractor is based on RoBERTa in which a word is tokenized into multiple tokens. As a result, there could be a case that some tokens of a single word are classified as a keyword while other tokens are determined as a non-keyword.

The first problem is solved by adopting a simple heuristic, which is to consider the token with the highest probability as a keyword if all tokens are predicted as non-keywords. That is, if all $y_i$'s are non-keywords, i.e., $p_i^T < p_i^F$ for $1 \leq i \leq n$, then $y_q$ is forced to be a keyword where $q$ is chosen by

$$q = \underset{i \in \{1, \dots, n\}}{\arg\max} \; p_i^T.$$

As a result, at least one token survives as a keyword.

This paper proposes two post-processings of *reinforcement* and *exclusion* as a solution to the second problem. Assume that a word $w_i$ is composed of $k$ tokens of $t_{i1}, \dots, t_{ik}$ and

the keyword label for $w_i$ is provided as $[y_{i1}, \ldots, y_{ik}]$, where $y_{ij} = f(t_{ij}; \mathbf{X})$. Some $y_{ij}$'s are keyword and the remaining are non-keyword since the classifications for $t_{ij}$'s are performed independently. The *reinforcement* post-processing enforces $w_i$ to be a keyword even when some $y_{ij}$'s in $w_i$ are non-keyword. Algorithm 1 and Figure 2a show how this post-processing works in detail. If there is at least one token of which label is keyword, the word $w_i$ is forced to be keyword. Thus, in lines 3 and 4, the algorithm checks if there is any token $t_{ij}$ satisfying $f(t_{ij}; \mathbf{X}) = \text{keyword}$. On the other hand, the *exclusion* post-processing enforces $w_i$ to be a keyword only when all $y_{ij}$'s are keyword. Algorithm 2 and Figure 2b show the flow of the *exclusion* post-processing. Similar to Algorithm 1, this algorithm checks if there is any token $t_{ij}$ satisfying $f(t_{ij}; \mathbf{X}) = \text{non-keyword}$ in lines 3 and 4. Thus, if there is at least one non-keyword token, $w_i$ also becomes non-keyword.

---

**Algorithm 1** The algorithm of reinforcement post-processing

---

**input:** a word $w_i = [t_{i1}, \ldots, t_{ik}]$
**output:** keyword or non-keyword

1: *is_keyword* ← non-keyword
2: **for** $1 \leq j \leq k$ **do**
3:     **if** $f(t_{ij}; \mathbf{X}) = \text{keyword}$ **then**
4:         *is_keyword* ← keyword
5:     **end if**
6: **end for**
7: **return** *is_keyword*

---

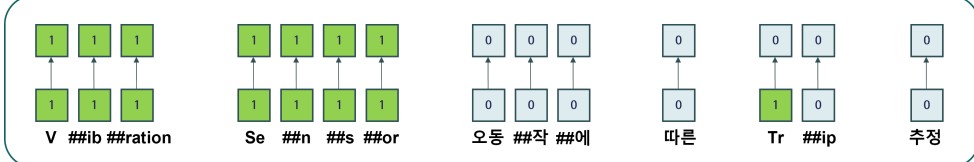

(**a**) Reinforcement post-processing

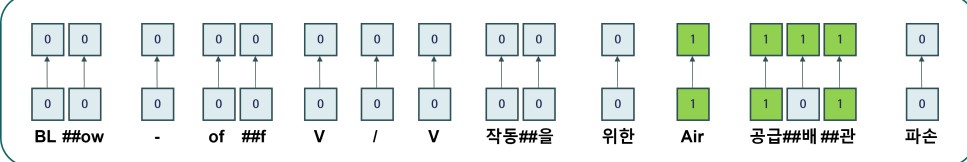

(**b**) Exclusive post-processing

**Figure 2.** The examples of two types of post-processing. The example in (**a**) is "Trip estimation due to vibration sensor malfunction", and the example in (**b**) is "Damage of air supply pipe for blow-off V/V operation". If the label is 1, it is classified as a keyword, and, if the label is 0, it is classified as a non-keyword. Words are separated through wide spacing.

---

**Algorithm 2** The algorithm of exclusion post-processing

---

**input:** a word $w_i = [t_{i1}, \ldots, t_{ik}]$
**output:** keyword or non-keyword

1: *is_keyword* ← keyword
2: **for** $1 \leq j \leq k$ **do**
3:     **if** $f(t_{ij}; \mathbf{X}) = \text{non-keyword}$ **then**
4:         *is_keyword* ← non-keyword
5:     **end if**
6: **end for**
7: **return** *is_keyword*

---

## 4. Experiments

### 4.1. Dataset

To prove that the proposed keyword extractor solves the problems that easily occur in keyword extraction supervised with small number of documents, a corpus named as *Korean Power Plant Outage Reports* is adopted. Every report in the corpus has three sections of 'Failure Detail', 'Cause', and 'Measurement'. One dataset is made from one section. As a result, three datasets of *failure detail*, *cause*, and *measurement* are finally prepared from the corpus. Table 1 shows an example of the data.

**Table 1.** An example of Korean Power Plant Outage Reports dataset.

| Dataset | Document | Object | Phenomenon |
|---|---|---|---|
| *failure detail* | ST 복수기 냉각수(해수) 펌프 고장으로 인한 ST Trip <br> *ST Trip due to ST condenser coolant (sea water) pump failure* | ST <br> *ST* | Trip <br> *Trip* |
| *cause* | m ST 복수기 냉각수(해수)펌프 고장에 의한 냉각수 공급중단으로 복수기 진공도 하락 후 ST 진동 상승하여 Trip <br> *m ST condenser vacuum level drops due to cooling water supply interruption due to failure of the ST condenser coolant (seawater) pump, then ST vibration increases and trips* | ST 복수기 냉각수(해수) 펌프 <br> *ST condenser coolant (seawater) pump* | 고장 <br> *failure* |
| *measurement* | 냉각수(해수) 공급 펌프 Shaft 등 신품설치 추진 <br> *Promotion of installing new products such as cooling water (sea water) supply pump shaft* | 냉각수(해수) 공급 펌프 Shaft 등 신품설치 추진 <br> *cooling water (sea water) supply pump* | 신품 설치 <br> *installing new products* |

The keywords in a data instance are tagged from two points of view. One point of view is about failure object and the other is about failure phenomenon. Thus, every data instance has two types of keywords: *object* and *phenomenon*. In the dataset of *failure detail*, all instances have only one *object* keyword and one *phenomenon* keyword. On the other hand, in *cause* and *measurement*, the instances have three or fewer keywords for both *object* and *phenomenon* types.

The number of data instances in each dataset is 300, and the average input lengths of *failure detail*, *cause*, and *measurement* are 50.05, 129.59, and 87.56, respectively. The average numbers of the keywords in the datasets are 1.00, 1.10, and 1.23. Since the number of data instances is not large, the five-fold cross-validation is used to validate the proposed keyword extractor.

### 4.2. Implementation Details

The hyper-parameters for RoBERTa follow the settings of `klue/bert-base` (https://huggingface.co/klue/bert-base, accessed on 18 October 2023). The embedding size of hidden state is 768, the number of attention heads and the number of layers are twelve, and the size of vocabulary is 32,000. The cross-entropy is adopted for loss function to train the proposed keyword extractor, and AdamW is adopted as an optimizer with a learning rate of $2 \times 10^{-8}$ $\beta_1$, $\beta_2$, and $\epsilon$ for AdamW are set as 0.9, 0.999, and $1 \times 10^{-8}$, respectively.

In order to implement the proposed keyword extractor, Python 3.9.17, PyTorch 2.0, and Transformers 4.33.1 are used. The environment for training the keyword extractor is a PC with an RTX 3090-Ti GPU, an Intel i9-10980XE CPU, and 256GB RAM. The maximum sequence length of an input is 256, and the size of mini-batch is 16.

### 4.3. Evaluation Metrics

The performance of the proposed keyword extractor is measured with well-known automatic metrics: F1-Score, Micro (micro-accuracy), and Macro (macro-accuracy). F1-Score calculates the harmonic mean of the precision and the recall, while Micro calculates the token-level accuracy and Macro calculates the instance-level accuracy.

### 4.4. Baselines

The proposed keyword extractor is compared with the following baselines:

- VR: a vanilla RoBERTa model with a basic sequence labeling.
- BIO: a sequence labeling RoBERTa model with the BIO-tags.
- SQUAD: a RoBERTa model with SQuAD tags used in SQuAD dataset [41].
- TNT-KID: a modified transformer-based keyword extractor that additionally incorporates part-of-speech information [10].
- BART: a transformer-based generative language model.

VR, BIO, and SQUAD are all RoBERTa models, while BART is a standard BART [26]. However, VR, BIO, and SQUAD are differently trained because the instances are labeled in different schemes. In VR, all token labels are assumed to be independent from one another. Thus, no word-level information is used by VR. On the other hand, the keywords are labeled with the BIO-tags as in NER-tasks in BIO, and they are labeled with a starting index and an ending index as in the SQuAD dataset in SQUAD. Thus, both BIO and SQUAD are trained with some word-level information.

### 4.5. Experimental Results

Table 2 shows how the post-processing affects the performance of keyword extraction in the *object* keyword type of the *failure detail* dataset. In this table, REIN implies the reinforcement post-processing and EXC is the exclusion post-processing. For all metrics, EXC achieves higher performance than REIN consistently. This is because the keyword extraction performance of RoBERTa is sufficiently excellent, and thus predicting a few tokens of a word as `keyword` can be considered as noise. Therefore, *exclusion* post-processing is used for all the experiments below.

**Table 2.** Comparison of two post-processing methods of *reinforcement* (REIN) and *exclusion* (EXC). Scores in bold stand for the leadership among the metrics.

| Metric | F1-Score | Micro | Macro |
|:---:|:---:|:---:|:---:|
| REIN | 0.62 | 87.21 | 70.00 |
| EXC | **0.65** | **90.36** | **73.33** |

As explained above, the keywords are tagged for a single data instance from two different points of view. Thus, every instance has two types of keywords. As a result, there can be two ways to determine the keywords of a data instance. One way is to train one model to determine both types, and the other way is to train two distinct models for the two types. For the former, the keywords from both types are all regarded as keywords for a data instance. Table 3 compares these two ways. The *failure detail* dataset is used to measure the performances in this table. According to this table, 'Object-Model' trained only for *object* type achieves higher performance in *object*-type keyword extraction than 'Both-Model' trained to extract both types of keywords. Similarly, 'Phen.-Model' trained only for *phenomenon* type outperforms 'Both-Model' in *phenomenon*-type keyword extraction. Therefore, it can be inferred that the use of two distinct models is more effective.

**Table 3.** Comparison of two methods for extracting two types of keywords. 'Both-Model' implies a single model trained to extract both types of keywords, while 'Object-Model' is a model trained to extract the *object*-type keywords, and 'Phen.-Model' is that to extract the *phenomenon*-type keywords. Scores in bold stand for the leadership among the keyword types.

| Keyword Type | *Object* | | | *Phenomenon* | | |
|---|---|---|---|---|---|---|
| Metric | F1-Score | Micro | Macro | F1-Score | Micro | Macro |
| Both-Model | 0.56 | 85.25 | 68.33 | 0.75 | 87.59 | 71.67 |
| Object-Model | **0.65** | **90.36** | **73.33** | - | - | - |
| Phen.-Model | - | - | - | **0.83** | **94.57** | **78.33** |

Table 4 reports how superior the proposed keyword extractor is to the baselines. The proposed keyword extractor outperforms all baselines in all datasets. The performance differences are all statistically significant with $p$-value $< 0.05$. In particular, TNT-KID and BART show extremely low performances. This is because the number of instances in the datasets is insufficient to newly train the modified transformer or fine-tune BART. That is, BART is ineffective when only small datasets are available. VR, BIO, and SQUAD are differentiated by the label-tagging scheme. SQUAD and BIO have to learn the order of B-tags and I-tags or the keyword positions. Since there are not many data instances to train, learning such information is more difficult than independent keyword prediction. This is why VR is better than both SQUAD and BIO in this table.

**Table 4.** The empirical comparison of the proposed keyword extractor with its baselines. Scores in bold stand for the leadership among the models.

| Keyword Type | Object | | | Phenomenon | | |
|---|---|---|---|---|---|---|
| **Metric** | **F1-Score** | **Micro** | **Macro** | **F1-Score** | **Micro** | **Macro** |
| **failure detail** | | | | | | |
| VR | 0.64 | 85.23 | 61.67 | 0.77 | 88.65 | 61.67 |
| BIO | 0.53 | 80.67 | 45.00 | 0.72 | 83.98 | 53.33 |
| SQUAD | 0.59 | 81.36 | 51.67 | 0.75 | 85.25 | 51.67 |
| TNT-KID | 0.13 | 23.58 | 11.35 | 0.11 | 20.09 | 10.13 |
| BART | 0.09 | - | - | 0.08 | - | - |
| Proposed | **0.65** | **90.36** | **73.33** | **0.83** | **94.57** | **78.33** |
| **cause** | | | | | | |
| VR | 0.56 | 83.17 | 45.00 | 0.47 | 87.08 | 41.67 |
| BIO | 0.53 | 81.15 | 43.33 | 0.41 | 85.77 | 36.67 |
| SQUAD | 0.50 | 80.25 | 41.67 | 0.35 | 84.15 | 31.67 |
| TNT-KID | 0.08 | 18.25 | 09.15 | 0.07 | 16.21 | 08.79 |
| BART | 0.05 | - | - | 0.04 | - | - |
| Proposed | **0.62** | **93.84** | **53.33** | **0.58** | **97.74** | **61.67** |
| **measurement** | | | | | | |
| VR | 0.36 | 83.15 | 33.33 | 0.38 | 87.20 | 33.33 |
| BIO | 0.34 | 82.51 | 33.33 | 0.37 | 82.77 | 35.00 |
| SQUAD | 0.33 | 80.16 | 25.00 | 0.36 | 82.15 | 23.33 |
| TNT-KID | 0.07 | 16.97 | 08.46 | 0.07 | 15.28 | 08.63 |
| BART | 0.03 | - | - | 0.04 | - | - |
| Proposed | **0.42** | **92.85** | **43.33** | **0.43** | **97.04** | **45.00** |

The proposed keyword extractor achieves the best performance in all datasets, but Macro-accuracy in *failure detail* is higher than in *cause* and *measurement*. This is because there is only one golden keyword in *failure detail*, but up to three keywords appear in *cause* and

*measurement*. Thus, keyword extraction for *failure detail* is easier than that for other datasets. In addition, SQuAD-tagged classification is intrinsically more difficult than BIO-tagged classification since a classifier has to predict both the starting and ending positions in a SQuAD-tagged dataset, but it determines only whether each word is a keyword or not in a BIO-tagged dataset. Thus, when multiple keywords should be found as in *cause* and *measurement* datasets, Macro-accuracy of SQUAD is lower than that of BIO.

### 4.6. Ablation Study

In Section 3.2, we have discussed the heuristics for two problems of learning with a small number of data instances. Let 'ONE' denote the first heuristic that remains at least one keyword. Then, Table 5 shows how effective each heuristic is as an ablation study in *failure detail*. According to Table 2, 'EXC' is superior to 'REIN'. Thus, only 'EXC' is considered for the second heuristic. Whenever each heuristic is removed, the performances keep going down. When 'EXC' is excluded, Micro- and Macro-Accuracy drop by 1.5 and 6.66, respectively. Moreover, when 'ONE' is removed, even F1-Score decreases. In a word, both 'ONE' and 'EXC' are effective in learning a keyword extractor with a small number of data instances.

**Table 5.** Ablation study on the post-processing methods. Scores in bold stand for the leadership among the metrics.

| Metric | F1-Score | Micro | Macro |
|:---:|:---:|:---:|:---:|
| Proposed | **0.65** | **90.36** | **73.33** |
| – EXC | 0.65 | 88.86 | 66.67 |
| – ONE | 0.64 | 85.23 | 61.67 |

## 5. Conclusions

This paper has proposed a novel keyword extractor trained with an extremely small dataset. Supervised keyword extraction can be divided into classification-based approaches and generation-based approaches. The classification-based approach is formulated as sequence labeling and finds out the keywords of a document by classifying whether each token in the document is a keyword. In contrast, the generation-based approach abstractly generates keywords. Therefore, it is able to generate meaningful keywords that do not appear in the document, but it is based on generative language models, and the generative language models usually require a large amount of data. If a generative language model is trained with a small dataset, it is easy to underfit to the dataset, which leads to poor performance.

The proposed keyword extractor takes the classification-based approach due to the small size of available data. It consists of three components of RoBERTa, a keyword estimator, and a decision rule. The proposed keyword extractor suffers from two problems since it is a token-level classifier based on RoBERTa. The first problem occurs when all tokens in a document are classified as `non-keywords`, and the second problem is that a single word composed of plural tokens can be determined to be both a `keyword` and a `non-keyword` since the token-level classifications are made independently. Therefore, this paper has proposed two kinds of heuristics to solve them. The first heuristic to solve the first problem is to determine the word to which the token with the highest probability belongs as a keyword for the document. The second problem is solved by adopting two post-processing methods: reinforcement and exclusion. The reinforcement post-processing considers a word as a keyword when some tokens of the word are classified as keywords, while the exclusion post-processing determines a word as a keyword only when all tokens of the word are classified as keywords.

The effectiveness of the proposed keyword extractor has been shown through the experiments on Korean Power Plant Outage Reports data. According to the experimental results, the proposed keyword extractor outperforms significantly its baselines in Micro-Accuracy and Macro-Accuracy as well as F1-Score. These results prove that the proposed

keyword extractor reduces the problems of a RoBERTa-based keyword extractor trained with a small dataset. In addition, it has also been shown empirically that the exclusion post-processing is superior to the reinforcement post-processing.

One thing to note in the experiments is that BART showed a very low F1-Score. This is because the number of data was so small that BART could not be sufficiently trained. The small data size is also the reason why other RoBERTa-based baselines achieve lower performances than the vanilla RoBERTa. Through an ablation study, it was proven that both the two heuristics are helpful in training the keyword extractor with a small dataset.

Great attention has been paid recently to the generation-based approach in keyword extraction because it enables natural keywords to be generated compared to the classification-based approach [42,43]. As aforementioned above, this approach is difficult to use when the size of available data is small. However, some studies on few-shot learning with a pre-trained language model [44,45] and learning a pre-trained language model for a low-resource language [46,47] are in progress. Thus, it is our future work to combine the proposed keyword extractor with such few-shot learnings.

**Author Contributions:** Conceptualization, S.-E.K. and J.-B.L.; methodology, S.-E.K.; software, G.-M.P.; validation, J.-B.L. and G.-M.P.; formal analysis, S.-E.K.; resources, S.-B.P.; data curation, S.-M.S.; writing—original draft preparation, S.-E.K.; writing—review and editing, S.-B.P.; visualization, S.-E.K.; supervision, S.-B.P.; project administration, S.-B.P.; funding acquisition, S.-B.P. All authors have read and agreed to the published version of the manuscript.

**Funding:** This research was supported by Korea Electric Power Corporation (R22X005-06) and by the MSIT (Ministry of Science and ICT), Korea, under the ITRC (Information Technology Research Center) support program (IITP-2023-RS-2023-00258649) supervised by the IITP (Institute for Information & Communications Technology Planning & Evaluation).

**Data Availability Statement:** Data were obtained from Korea Electric Power Corporation, and the data are not publicly available.

**Conflicts of Interest:** The authors declare no conflict of interest.

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
