# Peer review of "RoBERTa-Based Keyword Extraction from Small Number of Korean Documents"

_electronics, doi:10.3390/electronics12224560_

Round 1
Reviewer 1 Report
Comments and Suggestions for Authors
Dear Authors,
Thank you for your interesting research in the field of keyword extraction. You did an excellent job of providing an overview of issues with actual examples and presenting experimental values as evidence. I will have only two general questions, which I hope will help your future Readers understand the text in more detail:
1. Page 3, line 100. What do you mean by the meaning of "stopword" and how is it then different from a keyword?
2. Page 6, line 218. “The number of data instances in each data set is 300” - it seems to me that this can be shown by one particular case. Can you briefly tell us how the output characteristics for comparing two methods will change if you increase or decrease the number of data instances?
I wish you success in your future research!
Reviewer 2 Report
Comments and Suggestions for Authors
This paper proposes a novel keyword extractor based on the classification approach. The proposed keyword extractor consists of RoBERTa, a keyword estimator, and a decision rule, where RoBERTa encodes an input document, the keyword estimator calculates the probability that each token in the document becomes a keyword, and the decision rule finally determines whether each token is a keyword using the probabilities. It is a well-structured paper with interesting results. However, it requires further improvements before publication.
(1) In the abstract, the author should highlight the specific problems to be solved in this study at the beginning, and then lead to the solutions. At present, the description is not clear. At the end of the abstract, the author can briefly summarize the research conclusions. The current expression is too redundant and should be deleted appropriately.
(2) In the introduction section, you should give the novelty and the contributions of your works.
(3) Proofread the paper carefully to improve it grammatically.
(4) Very brief literature is presented, try to update it with some latest references. The following may be consider. For example, https://doi.org/10.3390/rs15133402ï¼›https://doi.org/10.1093/burnst/tkad003;http://dx.doi.org/10.1145/3513263 and so on.
(5) In the Section 4 of 4. Experiments, how to determine parameters of used methods?
Comments on the Quality of English Language
Moderate editing of English language required
Reviewer 3 Report
Comments and Suggestions for Authors
After reading this article, I acknowledge your work, and there are still some shortcomings.
1. The Related Work lacks a detailed review of the generative models.
2. It is suggested that the authors give a specific analysis for a certain case.
3. There are limitations to your research. Would you consider applying your method to other languages?
4. The authors claim that reinforcement and exclusion could solve the second problem. I think the authors’ explanation is not enough. Please give a visual proof.
5. The proposed method lacks comparison with SOTA ones.
6. The proposed approach lacks originality and appears to be just the utilization of a large language model.

Comments on the Quality of English LanguageMinor editing of English language required.
Reviewer 4 Report
Comments and Suggestions for Authors
The paper analyzes a keyword extraction engine. The paper is poor of scientific contents. I suggest a rejection.
Strengths: starting model of RoBERTa keyword estimator.
Points of weakness:
Few details are provided about the testing dataset to understand in which way the extractor is useful.
It is no clear the advantages if compared with other extraction engines.
The paper is similar to a testing report and it is no clear the scientific innovation.
I think that the paper is suitable for a journal having topics in informatics and not in electronics.
Comments on the Quality of English LanguageGood
Round 2
Reviewer 2 Report
Comments and Suggestions for Authors
ok
Comments on the Quality of English Languageok
Reviewer 3 Report
Comments and Suggestions for Authors
All my concerns have been addressed in the revised version.
Reviewer 4 Report
Comments and Suggestions for Authors
About comment 3:
"This paper proposed the keyword extractor that can be used with the extremely small size of the data. In addition, it has been proven that improving the results through post processing is more efficient than designing a new keyword extractor."
Please improve the paper by "quantifying" the use of the "small size of data" by comparing with other keword extractor, and how the post processing is "more efficient than a new keyword extractor".
I think that a more deeply analysis could improve the paper orienting it on a a research goal.
Comments on the Quality of English Languagegood
Round 3
Reviewer 4 Report
Comments and Suggestions for Authors
The authors should add more references about language processing to improve the paper.
I suggest:
-https://doi.org/10.3390/data8010011
-https://doi.org/10.3390/fi13060145
- https://doi.org/10.3390/sym12111923
-https://doi.org/10.3390/app11198812
Comments on the Quality of English LanguageGood
